TECHNICAL RELEASE

# Atria: an ultra-fast and accurate trimmer for adapter and quality trimming

Jiacheng Chuan[1,2], Aiguo Zhou[1,3], Lawrence Richard Hale[2], Miao He[4] and Xiang Li[1,*]

1  Canadian Food Inspection Agency, Charlottetown, PE C1A5T1, Canada
2  University of Prince Edward Island, Department of Biology, Charlottetown, PE C1A4P3, Canada
3  South China Agricultural University, Guangdong Laboratory for Lingnan Modern Agriculture, Guangzhou 510642, China
4  Sun Yat-sen University, School of Life Sciences, Guangzhou, Guangdong, 510275, China

## ABSTRACT

With advances in next-generation sequencing, adapters attached to reads and low-quality bases directly and implicitly hinder downstream analysis. For example, they can produce false-positive single nucleotide polymorphisms (SNP), and generate fragmented assemblies. There is a need for a fast trimming algorithm to remove adapters precisely, especially in read tails with relatively low quality. Here, we present Atria, a trimming program that matches the adapters in paired reads and finds possible overlapped regions using a fast and carefully designed byte-based matching algorithm ($O(n)$ time with $O(1)$ space). Atria also implements multi-threading in both sequence processing and file compression and supports single-end reads. Compared with other trimmers, Atria performs favorably in various trimming and runtime benchmarks of both simulated and real data. We also provide a fast and lightweight byte-based matching algorithm, which can be used in various short-sequence matching applications, such as primer search and seed scanning before alignment.

**Subjects** Software and Workflows, Bioinformatics, Software Engineering

**Submitted:**   11 April 2021

\* Corresponding author. E-mail: sean.li@inspection.gc.ca

Preprint submitted at https://doi.org/10.1101/2021.09.07.459340

## STATEMENT OF NEED

Next generation sequencing (NGS) produces massive, high-resolution genome sequence data to facilitate various biological applications. Illumina paired-end sequencing can read a DNA fragment from both ends and generate accurate reads for downstream bioinformatics analysis, such as assembly, resequencing, transcriptome profiling, variant calling, epigenome profiling, chromatin interaction, and chromosomal rearrangements [1, 2].

In paired-end library preparation, adapter sequences are the technical sequences ligated to both sides of inserts, which are the DNA fragments of interest. Then, DNA molecules with adapters are sequenced from both ends of the inserts so paired-end reads are generated. If insert sizes of paired-end reads are less than the read lengths, inserts are reversely complementary, and adapters are sequenced after reading through the inserts (Figure 1). Thus, adapter contamination in the 3′ end must be removed before downstream analysis.

Adapters can be cleaned by searching adapter sequences and/or aligning paired reads (Figure 1). To date, some trimmers, such as AdapterRemoval (RRID:SCR_011834) [3], Trim Galore (RRID:SCR_011847) [4], and Trimmomatic (RRID:SCR_011848) [5], use both types of information to clean adapters. However, when the quality of sequencing reads decreases,

**Figure 1.** Overview of Atria workflow.

trimming processes employing both types of information are likely to give different trimming suggestions. Trimmers thus face a bottleneck when working on trimming adapters at accurate positions. Also, extremely short adapters at the low-quality 3′ end are sometimes difficult to detect. Thus, trade-offs between trimming truncated adapters, and retaining inserts intact, become necessary.

These two issues hinder trimmers from cleaning adapter sequences and leaving DNA inserts intact. To combat this, we present Atria (RRID:SCR_021313), an integrated trimming program for NGS data. Atria (RRID:SCR_021313) uses a fast byte-based matching algorithm to detect adapters and reverse complementary regions of paired reads, and integrates carefully designed decision rules to infer true adapter positions. Thus, Atria (RRID:SCR_021313) can trim extremely short adapter sequences at accurate positions, and not over-trim reads without adapters (Figure 1).

In addition to adapter trimming, Atria (RRID:SCR_021313) integrates trimming and filtering methods, such as consensus calling for overlapped regions, quality trimming, homopolymer trimming, N trimming, hard clipping from both ends, and read complexity filtration.

## IMPLEMENTATION

The adapter-finding algorithms used in Atria (RRID:SCR_021313) can be categorized in the following portions: DNA encoding, matching algorithm, matching and scoring, decision rules, consensus calling, quality trimming, and IO optimization (Figure 2).

## DNA ENCODING

The DNA encoding algorithm is developed based on BioSequences, a Julia (RRID:SCR_021666) package from BioJulia [6]. The original BioSequences package encodes DNA bases A, C, G, T as four-bit codes 0001, 0010, 0100, 1000, respectively. Extended codes are also supported, such as N (1111), S (0110), and gap (0000). DNA sequences are encoded

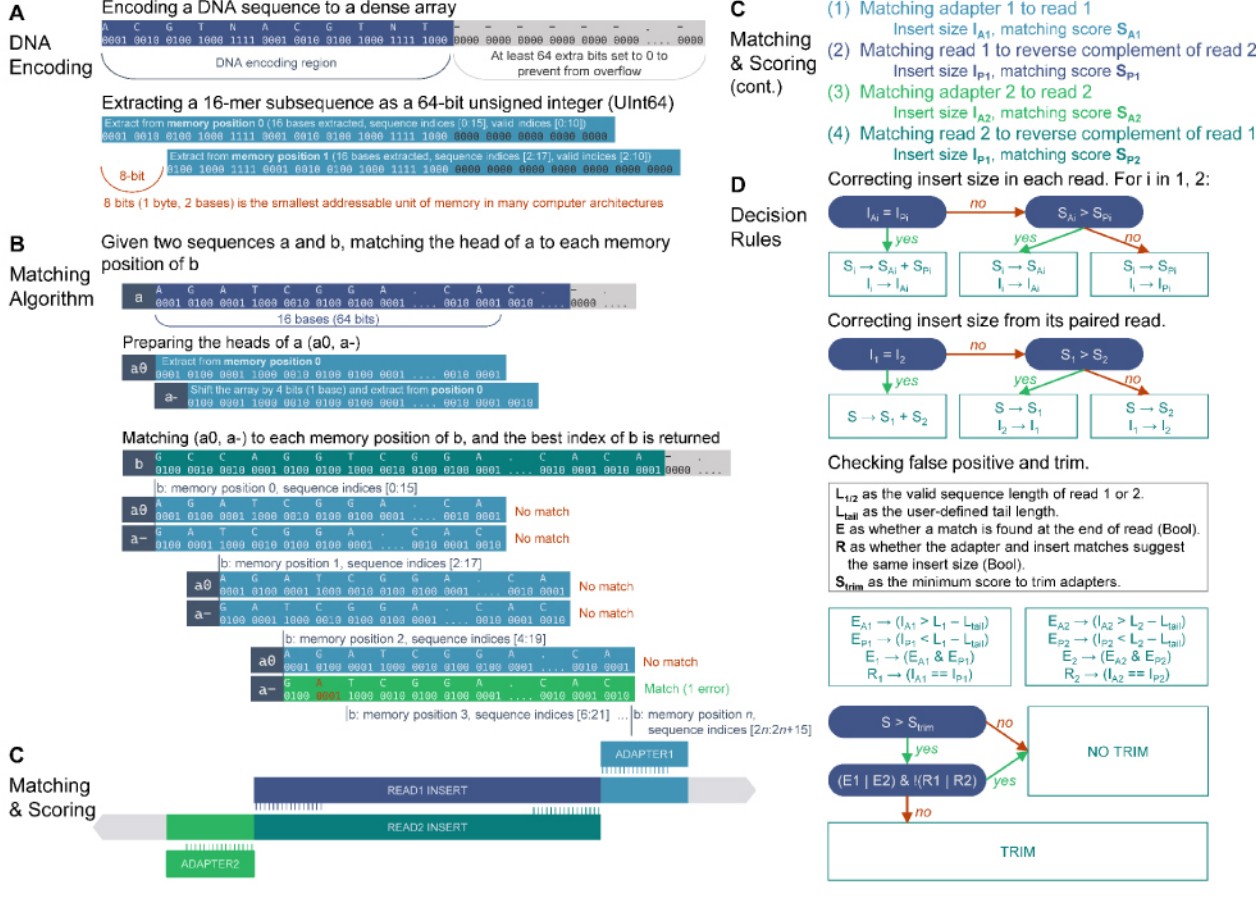

**Figure 2.** Adapter trimming algorithms.

and stored in a contiguous block of random-access memory (RAM) as a dense array of unsigned 64-bit integers (UInt64) (Figure 2A).

Atria (RRID:SCR_021313) makes use of the property of dense arrays to extract sequences as unsigned integers from available memory locations. When accessing the last several indices of a sequence, the extraction is illegal because operating systems do not allow the loading of data outside of sequence boundary. To solve the issue, Atria (RRID:SCR_021313) constructs a bit-safe sequence array, which elongates the sequence boundary by appending a UInt64 to the end of the original array, and setting all bits after the end of encoded DNA to 0 (Figure 2A).

It is noticeable that the smallest addressable unit of memory is 1 byte (8 bits) while each DNA is encoded in 4 bits, so only the even indices of sequences can be directly extracted (defining indices start from 0) (Figure 2A). The extraction from odd indices requires extra operations, which is avoidable in many scenarios of a well-designed algorithm.

We denote a UInt64 extracted from the memory position $n$ of sequence $a$ by $a_n$. $a_n$ is a 16-mer and represents the subsequence of $a$ indexed from $2n$ to $2n + 15$, which is denoted by $a[2n: 2n + 15]$ (Figure 2A).

## Matching algorithm

Given two sequences $a$ and $b$, we plan to match the 16-base-long head of $a$ to each index of $b$. However, only the even indices of $b$ can be extracted from memory without bitwise operations. Therefore, we prepared two UInt64 of $a$: $a_0$ and $a_-$. $a_0$ is the 16-mer UInt64 loaded from the position 0 of $a$, and $a_-$ can be computed from the following bitwise operations: $(a_0 \gg 4 \,|\, a_1 \ll 4)$. In this way, $a_0$ represents the subsequence of $a$ indexed from 0 to 15 ($a[0{:}15]$), and $a_-$ represents $a[1{:}16]$ (Figure 2B).

In this way, the problem of matching the 16-base-long head of $a$ to each index of $b$ is converted to the problem of matching two 16-mers, $a_0$, and $a_-$, to each addressable memory position of $b$. The latter requires less bitwise operations.

The number of mismatches $K$ is computed in the formula:

$$K_{an, bn} = 16 - count\_ones(an \ \& \ bn)$$

where *count_ones* counts the number of ones in the binary representation of the UInt64.

Let $k$ denote the user-defined number of mismatches allowed in the 16-mer comparison of UInt64 $a_n$ and $b_n$ ($k = 2$ by default). After matching $a_0$ and $a_-$ to each addressable memory position of $b$, if the minimum number of mismatches is not greater than $k$, the smallest index of $b$ of the minimum mismatches is reported.

Therefore, the complexity of the matching algorithm is $O(n)$ time with $O(1)$ space, so its speed is extremely fast. One limitation is that when computing the number of mismatches of $a_n$ and $b_n$, and if they have ambiguous bases in the same indices, the number of mismatches is underestimated. Another limitation is that the algorithm does not handle indels. Those limitations are compensated in the design of adapter matching, scoring, and decision rules.

## Matching and scoring

We implement four pairs of matching to utilize properties of paired-end reads thoroughly: (1) matching adapter 1 head to read 1, (2) matching adapter 2 head to read 2, (3) matching read 1 head to reverse complement of read 2 and (4) matching read 2 head to reverse complement of read 1 (Figure 2C). If the maximum number of bases matched of (1) and (2) is less than a user-defined cut-off (default is 9), (3) and (4) will be performed with a loosed $k$ ($= k_{\text{original}} + 1$). If the largest number of matched bases of the four matches is greater than the cut-off, and some matches do not meet the requirement, we will re-run those matches with a loosed $k$ ($= k_{\text{original}} + 3$) at the insert size indicated from the best match. If the new number of matched bases is greater than the cut-off, the old match will be discarded.

The scoring system measures the matching reliability of the whole 16-mer rather than each base. The Phred quality score $Q$ of each base is converted to the probability $P$ of that the corresponding base being correct using the formula:

$$P = 1 - 10^{(-\frac{Q}{10})}$$

Then, the average base quality $\overline{P}$ of 16-mer sub-sequence $a$ at the memory position $n$ is computed:

$$\overline{P_{a_n}} = \frac{1}{16} \sum_{i=2n}^{2n+15} P_{a[i]}$$

Notably, if the read quality is too low, it would imply an invalid match. However, in reality, invalid matches are filtered out by the kmer-based algorithm. To solve the discordance, we limit the lower bound of $\overline{P}$ to 0.75 manually.

The matching score $S$ between $a_n$ and $b_m$ is defined as:

$$S_{a_n,b_m} = count\_ones(a_n \ \& \ b_m) \cdot \overline{P_{a_n}} \cdot \overline{P_{b_m}}$$

where *count_ones* counts the number of ones in the binary representation of the UInt64. When sequence $a$ is a user-defined adapter, $\overline{P_a} = 1$ is used. Generally, the matching score $S$ is ranged from 0–16.

### Pseudocode 1: Matching and scoring

```
r1_pos_adpt, r1_nmatch_adpt = match(adapter1, r1, k)

r2_pos_adpt, r2_nmatch_adpt = match(adapter1, r1, k)

k_extra = max(r1_nmatch_adpt, r2_nmatch_adpt) < 9 ? 1 : 0

r1_pos_pe, r1_nmatch_pe = match(reverse_complement(r2), r1, k + k
    _extra)

r2_pos_pe, r2_nmatch_pe = match(reverse_complement(r1), r2, k + k
    _extra)

max_nmatch = max(r1_nmatch_adpt, r2_nmatch_adpt, r1_nmatch_pe, r2
    _nmatch_pe)

max_pos = corresponding position of max_nmatch

if max_nmatch > 9

  for matches with any nmatch < 9

    redo match with loosed  k = k + 3  at max_pos

    replace old results if nmatch > 9

r1_prob_adpt = average_16mer_quality(r1, r1_pos_adpt)

r2_prob_adpt = average_16mer_quality(r2, r2_pos_adpt)

r1_prob_head = average_16mer_quality(r1, 1)

r2_prob_head = average_16mer_quality(r2, 1)

r1_prob_pe = average_16mer_quality(r1, r1_pos_pe)

r2_prob_pe = average_16mer_quality(r2, r2_pos_pe)
```

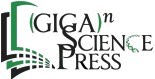

```
r*_prob_* = 0.75 if any r*_prob_* < 0.75

r1_score_adpt = r1_nmatch_adpt * r1_prob_adpt

r2_score_adpt = r2_nmatch_adpt * r2_prob_adpt

r1_score_pe = r1_nmatch_pe * r1_prob_pe * r2_prob_head

r2_score_pe = r2_nmatch_pe * r2_prob_pe * r1_prob_head
```

## Decision rules

This module infers correct adapter positions from the four pairs of matching described in the previous section. It is illustrated in Figure 2D. First, in each read, the adapter and paired-end matches are compared. The one with the higher matching score is chosen. If both matches support the same adapter position, the matching score of the read is the sum of adapter and paired-end matching scores. Then, the matches of the two paired-end reads are compared using the same strategy. If one read finds an ideal adapter (matching score >10 by default) while the other read is too short to check or the average base accuracy of its 16-mer is less than 0.6 (Phred $Q < 5$), both reads will be trimmed. If the matching score of a given read pair is less than 10 (by default), the read pair will not be trimmed.

Other read pairs will be further examined to reduce false positives, which are usually adapter matches at read tails. A read tail is defined as the last several bases (default is 12 bp) of each read. Reads are not trimmed if both statements are true: (1) in any paired read, the adapter is found at the tail, but the paired-end match is not; (2) in both paired reads, adapter and pair-end matches suggest different trimming positions.

Before the final trimming, one additional step is required for the accurate positioning of adapter sequences. The previous steps usually assume the read 1 and 2 have the same length of insert sizes, but indel in reads usually lead to over or under trim one base. To prevent this circumstance, Atria (RRID:SCR_021313) re-positions the adapter by matching one adjacent base with the first four base pairs of adapter sequences. The position of the highest number of bases matched is chosen to trim. This step is ignored when the inferred insert size is greater than the read length minus three because, in this situation, the adapter sequence is too short to check.

## Pseudocode 2: Decision rules

```
function correct_insert_size(pos1, score1, pos2, score2)

   if pos1 == pos2

     return pos1, score1 + score2

   else

     score = max(score1, score2)
```

```
            pos = corresponding pos of max score

            return pos, score

r1_pos, r1_score = correct_insert_size(r1_pos_adpt, r1_score_adpt,
    r1_pos_pe, r1_score_pe)

r2_pos, r2_score = correct_insert_size(r2_pos_adpt, r2_score_adpt,
    r2_pos_pe, r2_score_pe)

r12_pos, r12_score = correct_insert_size(r1_pos, r1_score, r2_pos,
    r2_score)

if r1_pos != r2_pos

    if r1_score > 10

      r2_prob = average_16mer_quality(r2, r1_pos)

          @goto ''trim'' if r2_prob < 0.6

    elseif r2_score > 10

      r1_prob = average_16mer_quality(r1, r2_pos)

          @goto ''trim'' if r1_prob < 0.6

function check_read_tail(read)

E_adpt = whether adapter found at read tail

E_pe = whether pair-end match found at read tail

E = E_adpt & E_pe # both matches in read tail

R = rx_pos_adpt == rx_pos_pe # adapter and pair-end match at same
    position

return E, R

E1, R1 = check_read_tail(r1)
```



```
E2, R2 = check_read_tail(r2)

E = E1 | E2 # at least one read matching in read tail

R = R1 | R2 # at least one read matching at the same position

is_false_positive = E & !R

if r12_score > trim_score & !is_false_positive

   @label ''trim''

   r1_pos_adjusted = adjacent_one_bp_check(r1, adapter1, r12_pos)

   r2_pos_adjusted = adjacent_one_bp_check(r2, adapter2, r12_pos)

      trim(r1, r1_pos_adjusted)

      trim(r2, r2_pos_adjusted)
```

## Consensus calling

In this module, the overlapped base pairs of read 1 and 2 are corrected to the corresponding bases with higher quality scores. It has three steps, prediction, assessment, and correction.

In the prediction step, Atria (RRID:SCR_021313) makes a preliminary estimate of whether a read pair contains an overlapped region. If adapters are trimmed and the remaining lengths of read 1 and 2 are the same, the prediction passes. If no adapter can be trimmed, two additional matching and scoring are required. The head of the reverse complement of read 2 is matched to read 1, and the head of the reverse complement of read 1 is matched to read 2. If the two matches reach a consensus, the prediction passes. Otherwise, the prediction fails and consensus calling is skipped.

In the assessment step, Atria (RRID:SCR_021313) compares the whole overlapped region using a similar matching algorithm, except that ambiguous bases (N, 1111) are converted to gaps (0000) before matching. If the ratio of mismatch is greater than a user-defined value (28% by default), the assessment fails, and consensus calling is skipped.

In the correction step, each base pair in the overlapped region is corrected to the corresponding base with the highest quality score.

## Quality trimming

Atria (RRID:SCR_021313) implements a traditional sliding window algorithm to remove the low-quality tail. The sliding window scans from the front of the read and computes the average Phred quality score of the sliding window. If the average quality is less than a given threshold, the read tail is removed.

## IO optimization

Atria (RRID:SCR_021313) spends more time on reading and writing than matching and trimming, so the key to reducing runtime is to optimize IO usage. Considering that a large amount of RAM is easily accessible nowadays, Atria (RRID:SCR_021313) trades increased RAM usage with decreased time. A large block of memory is allocated for reading input files, which is then wrapped and encoded to FASTQ objects parallelly using multi-threading. On the contrary, in the writing process, Atria (RRID:SCR_021313) unboxes and decodes FASTQ objects to string vectors in parallel and writes sequentially to files. In addition, pigz (parallel gzip) and pbzip2 (parallel bzip2) are called for compression and decompression when needed [7, 8]. Atria (RRID:SCR_021313) also support running with a single thread.

## Comparison to related work

### *Performance of adapter trimming on a simulated dataset*

We simulated 8.9 G bases with 100-bp paired-end reads from the *Arabidopsis thaliana* reference genome using the Skewer modified ART, a public NGS read simulator to allow adapters in the reads [9, 10]. The simulation profile was trained from a 101-bp paired-end public dataset SRR330569, and the 33-bp adapter pair used in read simulation is AGATCGGAAGAGCACACGTCTGAACTCCAGTCA and AGATCGGAAGAGCGTCGTGTAGGGAAAGAGTGT [11].

Atria (RRID:SCR_021313) v3.0.0 was benchmarked with cutting-edge and popular trimmers, including AdapterRemoval (RRID:SCR_011834) v2.3.1 [3], Skewer (RRID:SCR_001151) v0.2.2 [10], Fastp (RRID:SCR_016962) v0.21.0 [12], Ktrim v1.2.1 [13], Atropos v1.1.29 [14], SeqPurge v2012_12 [15], Trim Galore (RRID:SCR_011847) v0.6.5 [4] and Trimmomatic (RRID:SCR_011848) v0.39 [5]. Only adapter trimming was used, and other trimming and filtration were disabled. Detailed command line arguments are listed in Table S1, which is available in GigaDB that presents trimming speed on the 8.9 G bases 100 -bp paired-end simulated data [16]. Each trimming software was run on an idle Ubuntu 19.10 server with a 32-thread Intel i9-9960X central processing unit (CPU) at 3.10 GHz, 128 gigabyte (GB) DDR4-3200 RAM, and a 2 terabyte (TB) Samsung 970 EVO Solid State Drive (SSD) (sequential reads and writes up to 3.5 and 2.5 TB/s).

Trimming performance was evaluated based on the following metrics: positive predictive value (PPV), as the fraction of the number of correctly trimmed reads to all trimmed reads; sensitivity, as the fraction of the number of correctly trimmed reads to the reads with adapters; specificity, as the fraction of the number of untrimmed reads without adapters to all reads without adapters; and Matthew's correlation coefficient (MCC) measuring overall quality of pattern recognition, as:

$$\text{MCC} = \frac{\text{TP} \cdot \text{TN} - \text{FP} \cdot \text{FN}}{\sqrt{(\text{TP} + \text{FP})(\text{TP} + \text{FN})(\text{TN} + \text{FN})(\text{TN} + \text{FP})}}$$

where TP is the number of reads trimmed correctly, TN is the number of untrimmed reads without adapters, FP is the number of over-trimmed reads, and FN is the number of under-trimmed reads [3, 10].

The adapter trimming performance is shown in Table 1. AdapterRemoval (RRID:SCR_011834), Atria (RRID:SCR_021313) and Skewer (RRID:SCR_001151) were the top-class adapter trimmers in terms of MCC (99.61%, 99.51%, 99.44%, respectively) (Table 1). Fastp (RRID:SCR_016962) (98.92%) and Atropos (98.00%) were in the second tier (Table 1).



**Table 1.** Adapter trimming performance on the 8.9 G bases with 100-bp paired-end simulated data.

| Trimmer | PPV (%) | Sensitivity (%) | Specificity (%) | MCC (%) |
|---|---|---|---|---|
| Atria | 99.35 | 99.81 | 99.82 | 99.51 |
| AdapterRemoval | 99.42 | 99.94 | 99.83 | 99.61 |
| Atropos | 99.57 | 97.34 | 99.88 | 98.00 |
| Fastp | 98.73 | 99.58 | 99.61 | 98.92 |
| Ktrim | 91.51 | 85.85 | 98.84 | 87.84 |
| SeqPurge | 57.92 | 99.80 | 76.84 | 66.62 |
| Skewer | 99.58 | 99.53 | 99.88 | 99.44 |
| Trim Galore | 40.05 | 82.98 | 62.39 | 38.96 |
| Trimmomatic | 99.29 | 57.86 | 99.88 | 71.05 |

Ktrim obtained a good specificity (98.84%) but sacrificed its sensitivity (85.85%), and Trimmomatic (RRID:SCR_011848) achieved an exceptional specificity (99.88%) by trading off its sensitivity (57.86%) (Table 1).

To compare speed and efficiency, elapsed time (wall time) and average CPU consumption of each trimmer were recorded in different threading (1–32 threads) for uncompressed and gzip compressed data formats (Figure 3 and Table S1). Efficiency was defined as the fraction of processing speed to the percent of CPU utilized, so it was a better measurement, especially in CPU-intensive scenarios, such as running on a server with a job scheduling system or trimming multiple samples at the same time. Ktrim and Atria (RRID:SCR_021313) were two of the fastest trimmers in terms of speed and efficiency, from 1–16 threads (Figure 3 and Table S1). For uncompressed data, Trimmomatic (RRID:SCR_011848) was faster than Atria (RRID:SCR_021313) using 8–32 threads, but its real CPU usage was much greater than Atria (RRID:SCR_021313) (Figure 3 and Table S1). The speed and efficiency of AdapterRemoval (RRID:SCR_011834) and Skewer (RRID:SCR_001151) were generally 2–4 times less than Atria (RRID:SCR_021313), and Atropos was the slowest one (Figure 3 and Table S1). SeqPurge did not support the output of uncompressed data, so it was only tested in the compressed benchmark.

When trimming compressed data, the speed of AdapterRemoval (RRID:SCR_011834), Skewer (RRID:SCR_001151), Fastp (RRID:SCR_016962), Atropos and Trimmomatic (RRID:SCR_011848) remained constant when the number of threads increased from four to 32, because they failed to utilize more than four CPU in the IO process, while Atria (RRID:SCR_021313) and Trim Galore (RRID:SCR_011847) did not have the limitation (Figure 3 and Table S1). Atria (RRID:SCR_021313) was faster than Trim Galore (RRID:SCR_011847), and the efficiency of Atria (RRID:SCR_021313) was constantly two to three times greater than Trim Galore (RRID:SCR_011847) (Figure 3 and Table S1). SeqPurge showed strange speed curves; when assigning a single thread to SeqPurge, the average CPU usage was 300%, and the speed and average CPU usage dropped when assigning 8–32 threads (Figure 3 and Table S1). In addition, Ktrim did not support output compressed files, so we ignored it. Atria (RRID:SCR_021313) was usually the fastest trimmer when trimming compressed files.

### Detailed statistics of adapter trimming accuracy on a simulated dataset

The previous portion benchmarks on a whole dataset. This section evaluates trimming accuracy regarding different read properties, including adapter presence or absence, base error, and adapter length. To achieve the goal, Atria (RRID:SCR_021313) integrates a benchmarking toolkit for read simulation and trimming analysis.

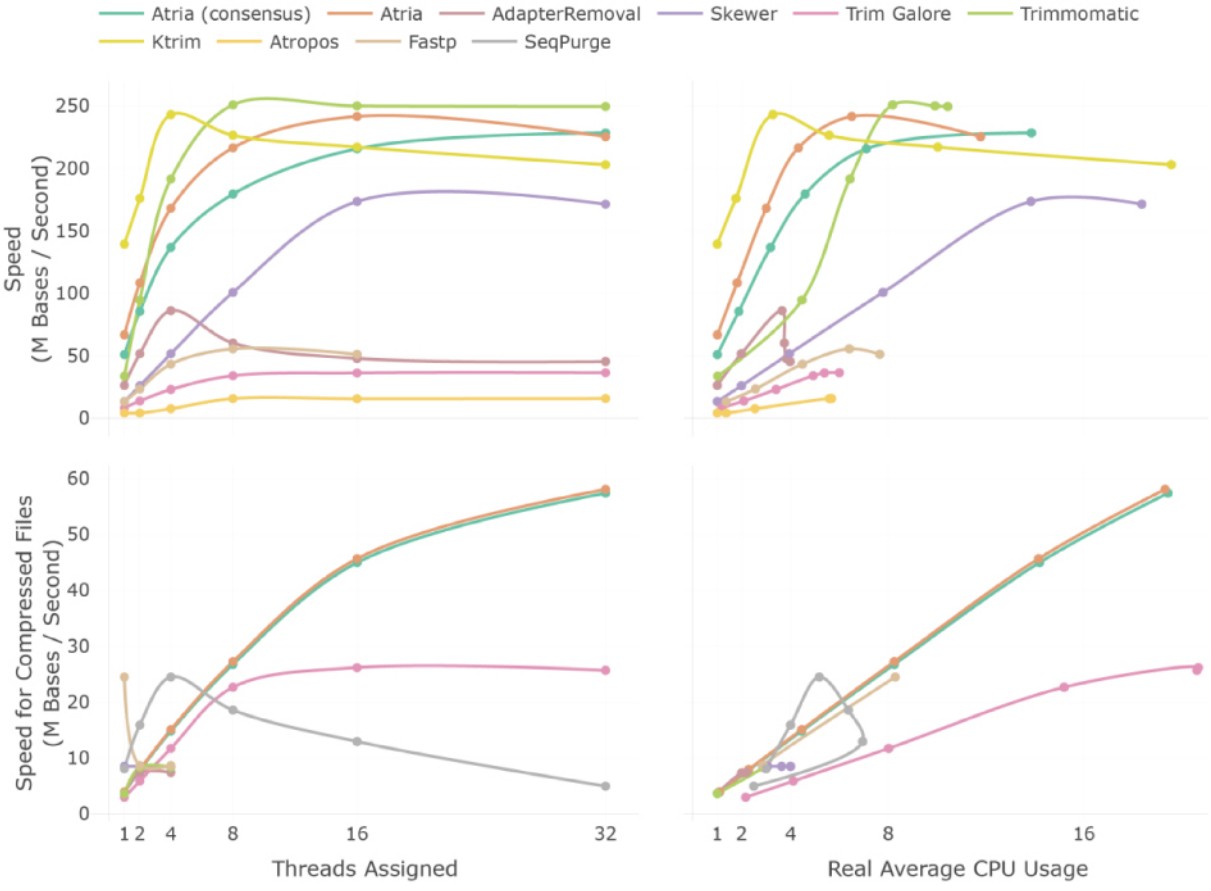

**Figure 3.** Benchmark of adapter-trimming speed for uncompressed and compressed files on different threading options. The 8.9 G bases simulating paired-end data with a 100 bp read length were trimmed in both uncompressed and compressed format using up to 32 threads. Speed is the ratio of the number of bases to elapsed time (wall time). SeqPurge does not support uncompressed outputs, so it is not shown in the uncompressed benchmark. In the trimming for compressed data, the speed of AdapterRemoval (RRID:SCR_011834), Skewer (RRID:SCR_001151), FastpFastp (RRID:SCR_016962), Atropos, and Trimmomatic (RRID:SCR_011848) kept constant when the number of threads increased from four to 32, so we only benchmark those trimmers using one, two, and four threads. Ktrim does not support output compressed files, so it is not shown in the compressed benchmark.

The read simulation method was inspired by the way in which sequencers read DNA. First, an original DNA fragment (insert) with a given original insert size is simulated, base by base. Adenine, thymine, cytosine, and guanine are randomly chosen repetitively. Then, the insert and adapter sequences are copied base by base with an error profile, which simulates the procedure of sequencing by synthesis. The error profile defines substitution rate, insertion rate, and deletion rate.

Twenty-one million read pairs were simulated with a uniform read length (100 bp), different error profiles, adapter length, and original insert sizes. The baseline error profile comprises a 0.1% substitution rate, 0.001% insertion rate, and 0.001% deletion rate, inspired by an Illumina error profile analysis [17]. We chose 1×, 2×, 3×, 4×, and 5× baseline error profiles; 16, 20, 24, 28, and 33 adapter lengths; and 66–120 even insert sizes. In this way, the reads with the least insert size have full lengths of adapters. The reads with 66–98 original insert sizes contain adapters, and the reads with 100–120 original insert sizes are free from adapter contamination, except for few reads with a 100-bp insert size containing indels. Therefore, in each condition combination, 30,000 read pairs were simulated to avoid

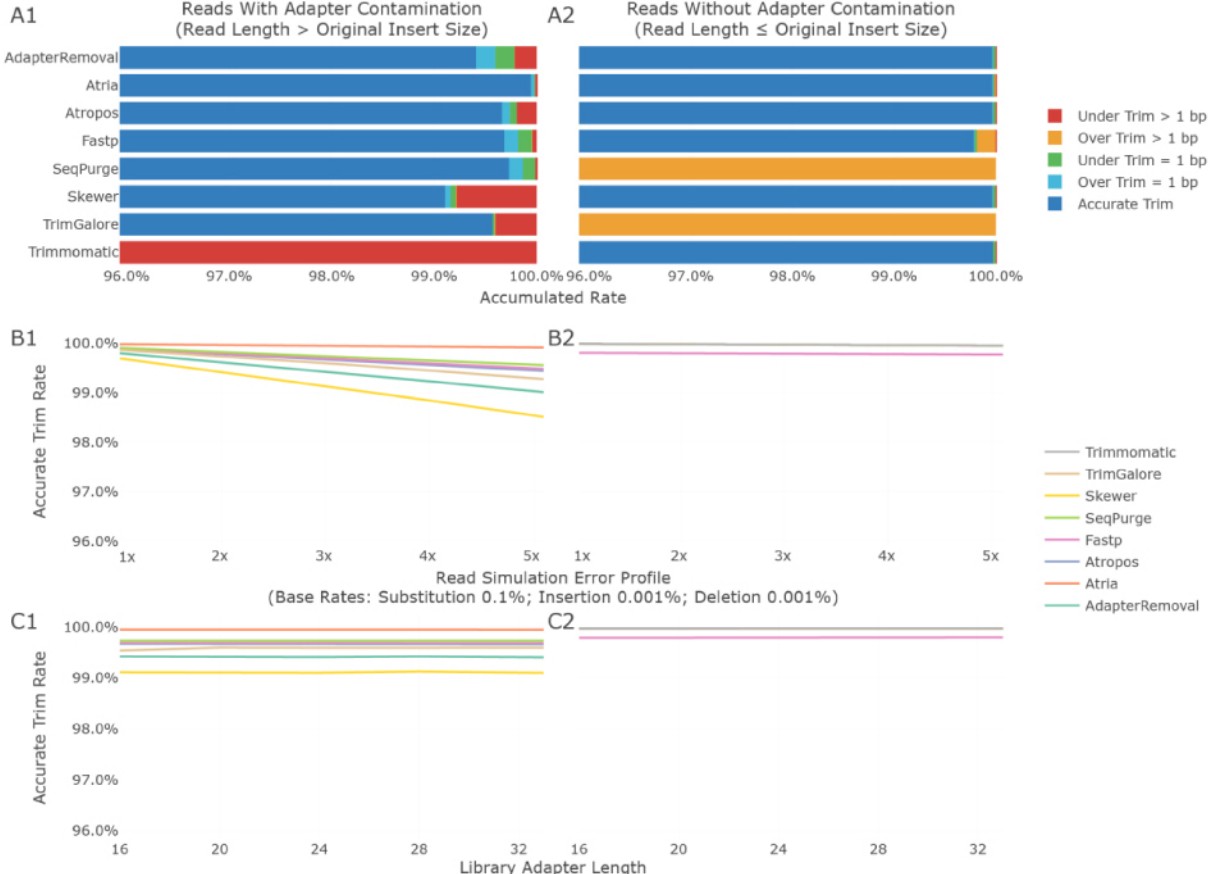

**Figure 4.** Adapter trimming accuracy on adapter presence and absence, different base errors, and adapter lengths. A1, B1, and C1 are statistics for reads with adapter contamination, while A2, B2, C2 are for reads without adapters. A1 and A2 show the accumulated rates of accurate trim, 1 bp over-trim, 1 bp under-trim, multiple bp over-trim, and multiple bp under-trim. In A1, the accuracy of Trimmomatic (RRID:SCR_011848) is 41.0%. In A2, the accuracy of SeqPurge is 78.8%, the accuracy of Trim Galore (RRID:SCR_011847) is 68.3%. B1 and B2 show the trimming accuracy of different error profiles. In B1, the accuracy of Trimmomatic (RRID:SCR_011848) drops from 41.9% to 40.1%. In B2, the accuracy of SeqPurge is 78.8%, and the accuracy of Trim Galore (RRID:SCR_011847) is 68.2–68.3%. C1 and C2 show the trimming accuracy of different adapter lengths. In C1, the accuracy of Trimmomatic (RRID:SCR_011848) is 0.0% at 16 bp adapter length, 50.7% to 51.6% at adapter lengths from 20–33 bp. In C2, the accuracy of SeqPurge ranges from 78.7% at 16 bp to 78.9% at 33 bp, and the accuracy of Trim Galore (RRID:SCR_011847) ranges from 68.2–68.3% from 16–33 bp.

random errors. Reads were trimmed with the same method described in the last section.

The average trimming performance among different conditions is shown in Figure 4A. When adapters are present, Atria (RRID:SCR_021313) trims 99.9% adapters accurately, and SeqPurge, Fastp (RRID:SCR_016962), and Atropos follow closely with an accuracy of 99.7% (Figure 4A1). When adapters are absent, AdapterRemoval (RRID:SCR_011834), Skewer (RRID:SCR_001151), Trimmomatic (RRID:SCR_011848), Atropos, and Atria (RRID:SCR_021313), successfully leave 100.0% reads intact, and Fastp (RRID:SCR_016962) falls behind with 99.8% accuracy (Figure 4A2).

Figure 4B illustrates the trimming accuracy on different read error profiles. When adapters are present, the accuracy of all trimmers drops as error rates increase (Figure 4B1). Atria (RRID:SCR_021313) keeps the highest accuracy from 100.0% to 99.9%, and is rarely affected by different error rates (Figure 4B1). The accuracy of SeqPurge, Fastp (RRID:SCR_016962), and Atropos decrease from 99.9% to 99.6%, 99.5%, and 99.4%,

respectively (Figure 4B1). With no adapter present in reads, the accuracy is influenced very little by error profiles (Figure 4B2), so the performance is similar to that shown in Figure 4A2. In addition, adapter lengths ranging from 16–33 bp are not relevant to the accuracy of most trimmers, including Atria (RRID:SCR_021313) (Figure 4C).

### Performance of adapter trimming on a real sequencing dataset

*RNA-seq paired-end dataset (SRR330569).* SRR330569 is a real RNA-seq dataset sequenced from *Drosophila simulans* with 5.46 G bases and 2 × 101-bp read length. It contains the 37-bp adapter sequences AGATCGGAAGAGCGGTTCAGCAGGAATGCCGAGACCG and AGATCGGAAGAGCGTCGTGTAGGGAAAGAGTGTAGAT in read 1 and read 2, respectively. Adapter trimming was performed by different trimmers without other trimming or filtering methods. Then, a sliding-window based quality trimming was performed to remove low-quality tails (sliding window size = 5 and average $Q$ score ≥ 15). The adapter-trimmed reads and adapter-and-quality-trimmed reads were mapped to the *D. simulans* genome version 2.02 from FlyBase using Hisat2 (RRID:SCR_015530) v2.2.1, respectively [18, 19]. Mapping statistics were collected using SAMTools (RRID:SCR_002105) Stat v1.10 [20]. Skewer (RRID:SCR_001151) did consensus calling after adapter trimming, and no option was provided to disable it. To achieve benchmark parity, Skewer (RRID:SCR_001151) was compared to Atria (RRID:SCR_021313) with consensus calling enabled, and other trimmers were compared to Atria (RRID:SCR_021313) without consensus calling. Time trimming was recorded in accordance with a common scenario: inputs were gzip-compressed and trimmed with eight threads, and outputs were also gzip-compressed to reduce massive disk use. All tested trimmers worked in the scenario except that Ktrim could not output gzip files (Table 2).

Atria (RRID:SCR_021313) was the fastest program to process and output compressed data in terms of wall time (Table 2). It also achieved the highest number of reads mapped and paired, and the highest percentage of properly paired reads with or without quality trimming. Usually, higher base mapping is accompanied by a higher error rate in the mapping process, so it is important to interpret the two metrics together. Atria (RRID:SCR_021313) had the lowest mapping error rate of 8.1833‰ and the forth highest number of bases mapped (Table 2). The trimmers (AdapterRemoval (RRID:SCR_011834), Fastp (RRID:SCR_016962), and Atropos) with the highest three error rates have the highest number of bases mapped (Table 2). Our program was usually more than 5% better than other trimmers for data without quality trimming (Table 2). Mapping statistics of data without quality trimming were usually worse than with quality trimming, except for Atria (RRID:SCR_021313). Specifically, the properly paired rates of other trimmers without quality trimming were 0.5–4% less than with quality trimming (Table 2). Quality trimming also increased the number of mapped and paired reads and reduced the number of unmapped reads (Table 2).

*Genome-wide human cell-free DNA dataset (ERR4695159).* Plasma cell-free DNA is usually short in length [21], and trimming is extremely important in medical diagnosis. Here, we chose a human genome-wide cell-free DNA dataset ERR4695159. It has 8.4 G bases with 2 × 150 bp read length with 33 bp adapter sequences AGATCGGAAGAGCACACGTCTGAACTCCAGTCA in read 1 and AGATCGGAAGAGCGTCGTGTAGGGAAAGAGTGT in read 2. The benchmark workflow was the

**Table 2.** Performance of trimmers on real data.

| Metric | Trimming and consensus | | | Trimming only | | | | | | |
| | Atria | Skewer | Atria | AR | Atropos | Fastp | Ktrim* | SeqPurge | Trim Galore | Trimmomatic |
|---|---|---|---|---|---|---|---|---|---|---|
| **Low-quality dataset (SRR330569, RNA, Hisat2 mapping)** | | | | | | | | | | |
| Elapsed time (min:sec)* | 2:38 | 9:19 | **2:32** | 11:29 | 10:08 | 9:17 | 1:34 + GZ | 3:53 | <u>3:39</u> | 9:38 |
| **No quality trimming** | | | | | | | | | | |
| Reads mapped and paired | 26,126,804 | 24,694,330 | **25,781,268** | <u>24,559,060</u> | 24,505,656 | 24,545,646 | 24,196,658 | 24,240,072 | 24,046,542 | 22,797,620 |
| Reads unmapped | 27,276,761 | 28,254,804 | **27,379,299** | <u>28,338,455</u> | 28,410,022 | 28,350,294 | 28,747,248 | 28,591,442 | 28,647,873 | 29,649,287 |
| Properly paired reads (%) | 48.3 | 45.6 | **47.6** | <u>45.3</u> | 45.2 | <u>45.3</u> | 44.5 | 44.7 | 44.2 | 38.3 |
| Base mapped (cigar) | 2,387,212,225 | 2,354,164,041 | 2,322,436,545 | **2,346,791,204** | 2,341,355,822 | <u>2,344,847,438</u> | 2,316,915,673 | 2,304,776,514 | 2,317,321,510 | 2,237,846,534 |
| Error rate (‰) | 7.3952 | 9.5897 | **8.1833** | 9.8902 | 9.8536 | 9.8683 | 9.7920 | 9.7904 | 9.6994 | <u>9.3106</u> |
| **With quality trimming** | | | | | | | | | | |
| Reads mapped and paired | 25,942,092 | 25,787,464 | **25,728,206** | 25,721,788 | 25,714,956 | <u>25,725,480</u> | 25,473,670 | 25,364,392 | 25,654,498 | 24,744,754 |
| Reads unmapped | 27,245,720 | 27,364,827 | <u>27,361,655</u> | 27,369,773 | 27,373,854 | **27,360,527** | 27,556,820 | 27,736,292 | 27,400,932 | 28,064,739 |
| Properly paired reads (%) | 47.9 | 47.6 | **47.5** | **47.5** | **47.5** | **47.5** | 46.9 | 46.8 | 47.3 | 42.3 |
| Base mapped (cigar) | 2,317,238,536 | 2,316,981,456 | 2,302,740,463 | **2,304,584,269** | 2,304,325,743 | <u>2,304,437,244</u> | 2,292,034,762 | 2,263,465,110 | 2,297,815,439 | 2,246,076,618 |
| Error rate (‰) | 7.1114 | 7.7882 | <u>7.8902</u> | 7.9160 | 7.9141 | 7.9149 | 7.9059 | 7.8787 | 7.8921 | **7.5649** |
| **High-quality dataset (ERR4695159, cell-free DNA, Bowtie2 mapping)** | | | | | | | | | | |
| Elapsed time (min:sec)* | 3:08 | 11:34 | **3:03** | 13:48 | 13:41 | 11:29 | 1:41 + GZ | <u>4:05</u> | 4:34 | 11:44 |
| **No quality trimming** | | | | | | | | | | |
| Reads mapped and paired | 54,367,548 | 54,287,616 | <u>54,324,964</u> | 54,319,438 | 54,299,922 | 54,322,088 | 53,087,420 | **54,446,104** | 54,218,344 | 54,128,760 |
| Reads unmapped | 1,094,145 | 1,016,244 | 1,119,103 | <u>989,335</u> | 1,002,745 | **978,665** | 2,317,968 | 999,099 | 1,041,005 | 1,094,752 |
| Properly paired reads (%) | 96.8 | 96.7 | <u>96.7</u> | <u>96.7</u> | <u>96.7</u> | <u>96.7</u> | 94.1 | **97.0** | 96.4 | 88.6 |
| Base mapped (cigar) | 7,703,820,585 | 7,700,134,673 | 7,700,298,217 | <u>7,701,482,302</u> | 7,700,749,164 | 7,699,298,008 | 7,607,799,306 | 7,512,839,360 | 7,677,087,845 | **7,720,352,493** |
| Error rate (‰) | 3.3082 | 3.8388 | 3.8724 | 3.8771 | 3.8834 | <u>3.8564</u> | 4.3239 | **3.8007** | 3.9173 | 6.1984 |
| **With quality trimming** | | | | | | | | | | |
| Reads mapped and paired | 54,553,566 | 54,526,276 | 54,546,192 | 54,541,948 | 54,539,502 | <u>54,549,462</u> | 53,335,674 | **54,608,308** | 54,482,002 | 54,403,982 |
| Reads unmapped | 965,447 | 984,845 | 967,917 | 970,869 | 973,217 | **826,424** | 2,136,081 | <u>890,884</u> | 999,918 | 914,003 |
| Properly paired reads (%) | 97.0 | 97.0 | <u>97.0</u> | <u>97.0</u> | <u>97.0</u> | <u>97.0</u> | 94.4 | **97.1** | 96.8 | 89.0 |
| Base mapped (cigar) | 7,653,879,312 | 7,649,380,218 | 7,646,989,362 | 7,647,893,624 | <u>7,648,184,196</u> | 7,646,574,606 | 7,556,468,882 | 7,461,588,482 | 7,625,484,706 | **7,668,777,971** |
| Error rate (‰) | 2.9547 | 3.2535 | 3.2678 | 3.2698 | 3.2792 | <u>3.2634</u> | 3.7183 | **3.2117** | 3.3109 | 5.5798 |

*Note*: AR = AdapterRemoval. In the trimming-only benchmark, bold and underline formats indicate the first and second trimmers (including tie) in terms of each metric, respectively. *Elapsed time (wall time) is benchmarked based on trimming and output gzip files with 8 threads, except that Ktrim cannot output gzip files (marked with time + GZ).

same as the RNA-seq analysis, except that the clean reads were mapped to the human reference genome hg38 (GRCh38.p13) using Bowtie2 (RRID:SCR_016368) v2.3.5.1 [22].

Atria (RRID:SCR_021313) was also the fastest trimmer in the scenario (3 min 3 s) (Table 1). SeqPurge and Trim Galore (RRID:SCR_011847) finished the task in more than 4 minutes, while others spent more than 11 minutes (Table 2).

In adapter trimming-only statistics, SeqPurge had the highest mapped and paired reads (54,446,104) and the highest properly paired reads (97.0%) (Table 2). Atria (RRID:SCR_021313) followed with 54,324,964 mapped and paired reads. Atria (RRID:SCR_021313), AdapterRemoval (RRID:SCR_011834), Atropos, and Fastp (RRID:SCR_016962) all had 96.7% properly paired reads (Table 2).

With quality trimming, the overall performance increased, and properly paired reads were closer; SeqPurge had 97.1% properly paired reads, with Atria, AdapterRemoval, Atropos, and Fastp close behind at 97.0% (Table 2). Only 89.0% of reads were properly paired with Trimmomatic (Table 2).

## DISCUSSION

Atria (RRID:SCR_021313) performs favorably with other cutting-edge adapter trimmers in accuracy, robustness, speed, and efficiency. Its performance is ascribed to the byte-based matching algorithm. The design concept of the algorithm is to minimize any unnecessary CPU operations by taking advantage of the data structure of dense arrays.

Matrix-based algorithms, such as the Needleman–Wunsch algorithm and the Smith–Waterman algorithm, allocate and update a matrix and perform base-to-base comparison [23, 24]. They report every mismatch and gap between two sequences, while Atria (RRID:SCR_021313) skips this step since it is focused on the start positions of successful matches. Despite that, the matching algorithm used in Atria (RRID:SCR_021313) is able to identify mismatch loci when needed.

The byte-based matching algorithm is lightweight and designed for short sequence scanning. Each DNA is encoded in 4 bits and stores continuously in RAM. A subsequence can be extracted as an unsigned integer from a given memory position. For example, a 64-bit unsigned integer (UInt) represents a 16-mer, and a 128-bit UInt represents a 32-mer. The comparison between two subsequences is completed within the accumulator register, a CPU unit for arithmetic or logical operation. It does not require addressing or updating a scoring matrix from RAM. When comparing a short sequence, such as an adapter, to a long sequence, such as the read, the 16-mer of the short sequence is compared to every position of the long sequence. Hence, the byte-based matching algorithm has $O(n)$ expected time complexity and $O(1)$ space complexity in adapter matching, where $n$ is the length of the long sequence.

The algorithm also has its limitations. It only reports the number of matched bases and does not report the positions of mismatches, so it cannot be used solely for sequence alignment. Besides, the algorithm does not handle inserts and deletions (indels). However, the average indel rate of an Illumina library is $10^{-6}$ to $10^{-5}$ [17], and the low indel rate is almost negligible in real data analysis. In addition, Atria (RRID:SCR_021313) matches four pairs in different locations to compensate for the limitation. If one match fails because of an indel, other matches will suggest the real adapter positions.

In the runtime benchmark, we compared the performance of trimmers using extremely high CPU cores. Efficiency usually marginally decreased as CPU usage increased owing to



the trimmers' parallel implementation and the inevitable cost of multi-threading, such as task scheduling and context switching. In addition, IO could be the main bottleneck for most hard disk drives and some solid-state drives. Thus, if the system IO reaches a bottleneck, an efficiency plateau would be expected sooner.

## CONCLUSIONS

We introduce not only Atria (RRID:SCR_021313), a cutting-edge trimming software for sequence data, but also the ultra-fast and lightweight byte-based matching algorithm. The algorithm can be used in various short-sequence matching applications, such as primer search and seed scanning before alignment. Atria (RRID:SCR_021313) is implemented in Julia (RRID:SCR_021666), a programming language designed specifically for high performance.

## AVAILABILITY OF SOURCE CODE AND REQUIREMENTS

Project name: Atria
Project home page: https://github.com/cihga39871/Atria
Operating system(s): Linux, OSX
Programming language: Julia
Other requirements: Julia v1.4, Pigz v2.4 or higher, Pbzip v1.1.13 or higher
License: MIT
Research Resource Identification Initiative ID: SCR_021313

## DATA AVAILABILITY

The datasets SRR330569, and ERR4695159 analyzed during the current study are available in the Sequence Read Archive from the National Center for Biotechnology Information [11, 26].

The Atria source codes, releases, documents, and benchmark scripts can be downloaded from Atria's Github page [25]. Snapshots of the code are also available in the *GigaScience* GigaDB repository [16].

## DECLARATIONS
## LIST OF ABBREVIATIONS

CPU: Central processing unit; MCC: Matthew's correlation coefficient; NGS: next-generation sequencing; PPV: positive predictive value; RAM: random-access memory; TB: terabyte; UInt: unsigned integer; UInt64: unsigned 64-bit integer.

## ETHICAL APPROVAL

Not applicable.

## COMPETING INTERESTS

The authors declare that they have no competing interests.

## FUNDING

This study was partially funded by the interdepartmental funding of Genomics Research and Development Initiatives (GRDI), Canada to XL. The financial support of CFIA and University of Prince Edward Island to JC is greatly appreciated.



## AUTHORS' CONTRIBUTIONS

JC developed Atria software, performed benchmark experiments under the supervision of XL. Both XL and LH served as co-supervisors and participated in the design of the study. MH contributed to optimization of the algorithm. AZ participated in benchmark validation. JC, LH, and XL drafted the manuscript. All authors read and approved the final version of the manuscript.

## ACKNOWLEDGEMENTS

The technical assistance of Jingbai Nie is greatly acknowledged. The advice on algorithm optimization, encouragement, and support of Drs. Christian Lacroix and Stevan Springer to JC are greatly appreciated.

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
