## [Reviewer Report]

Comments on revised manuscriptThe manuscript been modified as suggested, I do not have any additional comments .

---

## [Reviewer Report]

Comments on revised manuscriptI do not doubt that Atria get better efficiency given that it used 16 bp heads of adapters rather than the full length of adapter sequences and skipped the step of matrix-based alignment. My concern will be in its accuracy with the use of the performance-enhancing treatment mentioned above. All the test mentioned in the previous review, including the accuracy of Atria with different lengths of adapter sequences, verification of software accuracy from different aspects and the result with real large-scale datasets, it's all about the accuracy of Atria from different perspectives. As for the accuracy of Atria with different lengths of adapter sequences, a test of simulated adapters with different lengths/base quality/sequence error and the detailed statistics of the number of bases trimmed/kept in adapter sequences after trimming is very appreciate.  It is not easy to assess accuracy in real data, but I think the number of bases clipped/matched/mismatched may be a more direct indicator than the alignment rate. Overlapping and differences between different trimming tools are necessary to consolidate Atria's promotion in accuracy. Concerning the datasets used in the paper, real data SRR7243169 with a running time of 8-32 seconds seems to be not convincing enough. Also, the statement ” two of them are very large (8.4G, 5.6G), equivalent to the datasets in the 1000 genome project within the size of 400M-6.2G.” could be a little misleading. Mean coverage of 10X(~30G) reads is commonly sequenced for SNP calling.  If Atria does not have sufficient advantages over similar software in accuracy, two issues hinder trimmers from cleaning adapter sequences as mentioned in the abstract are not solved by Atria. Besides, I think that concluding statements that are not supported by the content should not be in the paper. The state “Atria is also compatible with single-end data from Illumina platforms and provides basic support for long-read data from PacBio and Nanopore platforms.” seems untenable as all the tests in the paper were based on paired reads from Illumina.

---

## [Reviewer Report]

Comments on revised manuscriptThe author has made some minor modifications to the content in this manuscript. However, much more needs to be done, for example, the tests mentioned in the previous review, which, disappointingly, are still not included in this version. Atria is more of a specialized tool dealing with NGS data with short length of adapter sequences. With many other good trimmers available, the reliability of Atria needs to be rigorously verified before it can be widely used. For example, the accuracy of Atria with different lengths of adapter sequences, verification of software accuracy from different aspects and the efficiency with real large-scale datasets. These tests, I believe, are necessary to consolidate the conclusions presented in this paper. Besides, the lack of indel processing greatly limits the wide use of the byte-based matching algorithm. There are some other problems in the writing. For example, the pseudo-code is not written properly, concluding statements that are not supported by the content (L80) should not be in the paper, also there are ambiguous symbols (L107) . BUT, never mind, please focus on the major ones and try your best to make this paper good enough to be published. Thank you.

---

## [Reviewer Report]

Reviewer name and names of any other individual's who aided in reviewerXingyu LiaoDo you understand and agree to our policy of having open and named reviews, and having your review included with the published manuscript. (If no, please inform the editor that you cannot review this manuscript.)YesIs the language of sufficient quality?YesPlease add additional comments on language quality to clarify if neededIs there a clear statement of need explaining what problems the software is designed to solve and who the target audience is? YesAdditional CommentsIs the source code available, and has an appropriate Open Source Initiative license <a href="https://opensource.org/licenses" target="_blank">(https://opensource.org/licenses)</a> been assigned to the code?YesAdditional CommentsAs Open Source Software are there guidelines on how to contribute, report issues or seek support on the code?YesAdditional CommentsIs the code executable?Unable to testAdditional CommentsIs installation/deployment sufficiently outlined in the paper and documentation, and does it proceed as outlined?YesAdditional CommentsIs the documentation provided clear and user friendly?YesAdditional CommentsIs there a clearly-stated list of dependencies, and is the core functionality of the software documented to a satisfactory level?YesAdditional CommentsHave any claims of performance been sufficiently tested and compared to other commonly-used packages? NoAdditional CommentsAre there (ideally real world) examples demonstrating use of the software? NoAdditional CommentsIs automated testing used or are there manual steps described so that the functionality of the software can be verified?YesAdditional CommentsTitle: Atria: An Ultra-fast and Accurate Trimmer for Adapter and Quality Trimming. Manuscript Number: TR-20210401. Opinion: Author Should Prepare a Major Revision. In this paper, the authors proposed a trimming algorithm called Atria, which matches the adapters in paired-end reads and finds possible overlapped regions with a super-fast and carefully designed byte-based matching algorithm. Furthermore, Atria implements multi-threading in both sequence processing and file compression and support single-end reads. The proposed algorithm has some significance in both theory and practical application. However, I still have some questions to discuss with authors. The comments on the paper are as follows. (1) Major Comments: 1) The author highlights the fast and accurate characteristics of the proposed trimming algorithm in the title of the manuscript. However, the large amount of content in the manuscript and supplementary is to prove the advantages of the proposed algorithm in terms of speed, processing efficient, and utilization of CPU and RAM. The assessment of trimming accuracy is very limited, and it seems that only general statistics are given in Table 2 of the manuscript. I personally think that the alignment rate of reads (or the number of paired-end reads) before and after trimming is not a good proof of the accuracy of the trimming algorithm. What's more, judging from the experimental results in Table 2, the Atria algorithm does not have much advantage in accuracy compared to other methods. As the author stated in the abstract, sequence trimming is of great significance for SNP detection and sequence assembly. I very much hope to see Atria's optimization and promotion of these applications. 2) The datasets used in this study seem to be unrepresentative, and most of them can be trimmed within a few to ten seconds. The difference between a few seconds and a dozen seconds, I think most users will not care. To prove the significant advantages of the proposed algorithm in terms of efficiency, some large-scale datasets (such as several samples sequenced in the 1000 genome project) should be used. (2) Minor Comments: 1) The table2 display of line 562 is incomplete.
Any Additional Overall Comments to the AuthorTitle: Atria: An Ultra-fast and Accurate Trimmer for Adapter and Quality Trimming. Manuscript Number: TR-20210401. Opinion: Author Should Prepare a Major Revision. In this paper, the authors proposed a trimming algorithm called Atria, which matches the adapters in paired-end reads and finds possible overlapped regions with a super-fast and carefully designed byte-based matching algorithm. Furthermore, Atria implements multi-threading in both sequence processing and file compression and support single-end reads. The proposed algorithm has some significance in both theory and practical application. However, I still have some questions to discuss with authors. The comments on the paper are as follows. (1) Major Comments: 1) The author highlights the fast and accurate characteristics of the proposed trimming algorithm in the title of the manuscript. However, the large amount of content in the manuscript and supplementary is to prove the advantages of the proposed algorithm in terms of speed, processing efficient, and utilization of CPU and RAM. The assessment of trimming accuracy is very limited, and it seems that only general statistics are given in Table 2 of the manuscript. I personally think that the alignment rate of reads (or the number of paired-end reads) before and after trimming is not a good proof of the accuracy of the trimming algorithm. What's more, judging from the experimental results in Table 2, the Atria algorithm does not have much advantage in accuracy compared to other methods. As the author stated in the abstract, sequence trimming is of great significance for SNP detection and sequence assembly. I very much hope to see Atria's optimization and promotion of these applications. 2) The datasets used in this study seem to be unrepresentative, and most of them can be trimmed within a few to ten seconds. The difference between a few seconds and a dozen seconds, I think most users will not care. To prove the significant advantages of the proposed algorithm in terms of efficiency, some large-scale datasets (such as several samples sequenced in the 1000 genome project) should be used. (2) Minor Comments: 1) The table2 display of line 562 is incomplete.
RecommendationMajor Revisions

---

## [Reviewer Report]

Reviewer name and names of any other individual's who aided in reviewerlialunDo you understand and agree to our policy of having open and named reviews, and having your review included with the published manuscript. (If no, please inform the editor that you cannot review this manuscript.)YesIs the language of sufficient quality?YesPlease add additional comments on language quality to clarify if neededIs there a clear statement of need explaining what problems the software is designed to solve and who the target audience is? YesAdditional CommentsIs the source code available, and has an appropriate Open Source Initiative license <a href="https://opensource.org/licenses" target="_blank">(https://opensource.org/licenses)</a> been assigned to the code?NoAdditional CommentsThere is no license in the github repository.As Open Source Software are there guidelines on how to contribute, report issues or seek support on the code?YesAdditional CommentsGithub can be used to report issues or seek support on the codeIs the code executable?YesAdditional CommentsIs installation/deployment sufficiently outlined in the paper and documentation, and does it proceed as outlined?YesAdditional CommentsIs the documentation provided clear and user friendly?YesAdditional CommentsIs there a clearly-stated list of dependencies, and is the core functionality of the software documented to a satisfactory level?YesAdditional CommentsHave any claims of performance been sufficiently tested and compared to other commonly-used packages? YesAdditional CommentsAre there (ideally real world) examples demonstrating use of the software? YesAdditional CommentsIs automated testing used or are there manual steps described so that the functionality of the software can be verified?YesAdditional CommentsAny Additional Overall Comments to the AuthorThe paper describes an ultra-fast and accurate trimmer for adapter and quality trimming: Atria  and compare it to several published tools. The tool is demonstrated to work on sequencing data with competitive accuracy and efficiency compared with existing tools. There are concerns that should be addressed: 1． The performance comparisons listed in Table 2 show that Atria is not extremely impressive compared with existing tools with quality trimming in percentage of the properly paired reads and the number of unmapped reads. Also, there are no more features than existing tools like Fastp, which may limit the widespread use of this software. 2． IO could be the main bottleneck for most hard-disk drivers when performing adapter trimming for compressed input/output files. So, the wall time to run different tools is also a good measurement. I wonder whether there is a significant advantage in performance if the runtime benchmark is measured by wall time. 3． Can the algorithm deal with different lengths of adapter sequences? It would be good to test out the performance of the tools with increasing length of adapter sequence. 4． L79 states that Atria is compatible with single-end data from Pacbio and Nanopore platforms, but there is no corresponding data in the paper to support the statement. Besides, the limitations of the byte-based matching algorithm make it difficult to deal with Pacbio and Nanopore sequences with high insert and deletion rates. It is necessary to describe how to get rid of these limitations in sufficient detail if they have been overcome. 5． It may be better if the description of this algorithm is presented in pseudocode especially in the section of “Matching and scoring” and “Decision rules”. 6． L165-L168, I don't quite understand why an adapter is an ideal adapter when the matching score is bigger than 10? Also, why the read pair will not be trimmed when the matching score is less than 19? Are there any reasons for the authors to set these two parameters 10 and 19 respectively? In addition, it is necessary for the authors to demonstrate that the program is robust enough for different lengths of adapter sequences. 7． All symbols in the paper should be clearly identified, e.g., L115 a1, L121 8． L135,” Because the matching algorithm requires much less time, we implement four pairs of matching to utilize properties of paired-end reads thoroughly”. The causation here does not hold.RecommendationMinor Revisions